# Emergence of Salmon Gill Poxvirus

**DOI:** 10.3390/v14122701

**Published:** 2022-12-01

**Authors:** Haitham Tartor, Maria K. Dahle, Snorre Gulla, Simon C. Weli, Mona C. Gjessing

**Affiliations:** Norwegian Veterinary Institute, Elizabeth Stephansens vei 1, 1431 Ås, Norway

**Keywords:** respiratory disease, red blood cells, fish, circulatory disturbance, anaemia, epithelial barrier, live rigor, immune response, viral transmission

## Abstract

The Salmon gill poxvirus (SGPV) has emerged in recent years as the cause of an acute respiratory disease that can lead to high mortality in farmed Atlantic salmon presmolts, known as Salmon gill poxvirus disease. SGPV was first identified in Norway in the 1990s, and its large DNA genome, consisting of over 206 predicted protein-coding genes, was characterized in 2015. This review summarizes current knowledge relating to disease manifestation and its effects on the host immune system and describes dissemination of the virus. It also demonstrates how newly established molecular tools can help us to understand SGPV and its pathogenesis. Finally, we conclude and ask some burning questions that should be addressed in future research.

## 1. Introduction

The farming of Atlantic salmon (*Salmo salar* L., hereafter salmon) in Norway is a huge industry, with around 350 million animals put to sea in 2021 [1]. This industry is threatened by the high mortality rates of fish in both freshwater farms prior to sea-transfer (estimated to be around 130 million fish in 2021) [1] and post sea-transfer (estimated to be about 54 million fish in 2021). Although gill diseases are a considerable contributing factor to these losses, there is no legal requirement to report their diagnosis to the public authorities. A broad overview of the situation is therefore lacking. Gills are multifunctional organs responsible for respiratory gas-exchange, but they also perform crucial functions in ion regulation, excretion of waste products, and immunity [2,3,4]. Compromised gills will, therefore, not only affect respiration, but also other crucial physiological functions. In the 1990s, cases of acute respiratory disease associated with high mortalities in juvenile salmon were reported in Norwegian freshwater farms. A poxvirus was suspected as the aetiological agent, based on a transmission electron microscopy (TEM) investigation (O. B. Dale and A. Kvellestad, unpublished). However, some years passed before convincing TEM images showing poxvirus particles were demonstrated [5,6], and the full virus genome sequence was published in 2015 [6]. This led to the development of diagnostic tools (PCR and antibodybased analyses) and demonstration of gill epithelial cells as the primary target cell-type for SGPV. Further, it became clear that in some cases, SGPV infection could be associated with epithelial destruction and severe, acute respiratory disease with extensive blood cell breakdown visible in some fish (Figure 1). The following phase is characterized by different degrees of gill epithelial hyperplasia and signs of compromised mucosal defense in the gills (Figure 1). 

## 2. Clinical Manifestations—From Asymptomatic Infections to High Mortality Outbreaks

The outcomes of SGPV infection in salmon may range from an apparently healthy state to fish with severe respiratory distress, leading to acute mortalities with losses of up to 70% of the animals in the affected tank [6]. Acute disease cases are the easiest to identify and have only been reported from hatcheries, in addition to some cases seen directly after sea transfer [8]. Outbreaks typically start within a couple of weeks following stressful events, such as handling of the fish or production related activities causing noise or vibration [9]. Further, clinical onset is often strikingly synchronised and characterised by morbidity and respiratory distress which rapidly escalates. Mortality often lasts for about four to seven days in an affected tank before spread to neighbouring tanks [7]. The pattern of spread may, however, vary [9]. Moribund fish will commonly display increased opercular movement reflecting respiratory distress, and some animals may display severe muscular rigidity while still alive, sometimes referred to as “live rigor” [9]. Sub-clinical SGPV infections are, however, also common, and such cases are associated with low viral loads in the gills. There is now a strong body of evidence indicating that the SGPV load, clinical manifestation, and pathological changes in gills are closely linked [6,7,8,10,11]. The association of a high virus load and severity of the disease has also been reported in koi carp (*Cyprinus carpio*) infected with another poxvirus, carp edema virus (CEV), leading to koi sleepy disease (KSD) [12]. The SGPV infection may also be identified in salmon after their transfer to the sea, often in combination with other gill pathogens, commonly diagnosed as complex gill disease (CGD) [13,14]. All components of the epidemiological triad (i.e., the host, the agent, and their shared environment) will undoubtedly affect the outcome of an SGPV infection. While not yet fully understood, stress seems—from both farm observations and experimental trials utilising cortisol implants—to be an important contributing factor [11]. While significant knowledge gaps remain in terms of eliminating salmon gill poxvirus disease (SGPVD), the effects of the disease may reportedly be reduced by stopping feeding, increasing oxygen levels, and avoiding stressful practices [9,15].

### 2.1. Is SGPVD a Manifestation of a Dysregulated Host Response in Combination with Respiratory Collapse?

External inspection of salmon suffering from SGPVD is often without specific findings, although redness of the abdominal wall and fin-bases may be observed (Figure 2) and has been suggested as an early indication of SGPVD development [9]. Autopsy is also commonly without specific findings apart from occasional reports of moderately pale gills, swollen spleen, and an empty gut [6].

In salmon suffering from SGPVD, characteristic histopathological changes are always seen in the gills, and in some cases also in the spleen and kidney, in the form of accumulation of apparently intracellular, amorphous eosinophilic material, interpreted as erythrophagocytosis (i.e., phagocytic destruction of red blood cells). In clinical outbreaks, a large part of the gill respiratory surface is impacted [6]. Apoptosis of gill epithelial cells, confirmed by TUNEL-positive staining, is a hallmark of SGPVD. Apoptotic cells containing SGPV particles may also be observed using TEM (Figure 3). The degree of apoptosis is quantitatively linked to the viral load and disease severity [6,7,8,10,11]. As the apoptotic epithelial cells detach, a widespread adherence of the thin gill lamellae can be seen (Figure 4). In the sub-acute phase of the disease (usually fish having survived a SGPVD outbreak), the inter-lamellar space may become blocked by hyperplastic and hypertrophic gill epithelial cells (Figure 4). Infiltration of leucocytes is not a common feature in SGPVD outbreaks (i.e., in cases where SGPV is suspected to be the only agent causing gill disease), and neither are visible inclusion bodies in the infected cells as described in orthopoxvirus infections [16,17]. Apoptosis is a regulated form of cell death functioning to eliminate unwanted or damaged cells while causing minimal destruction of the surrounding tissues. The specificity of apoptosis is also well suited for the removal of cells identified by the host as virus-infected, and interactions between viral proteins and host apoptotic pathways largely determine the outcome of infections [18]. Virus-driven prevention or reduction of apoptotic responses are reported for several large DNA viruses, including poxviruses [19]. For SGPV, however, the shedding of apoptotic gill epithelial cells leads to the release of infective virus particles into the water, allowing the virus to find new hosts. Another exception to the general anti-apoptotic nature of most poxvirus infections is reported for parapoxvirus, where apoptosis of the infected monocytes leads to impaired antigen presentation [20]. In KSD, impaired respiratory and excretory function of the gills, combined with osmotic, ionic, and acid-base disruption, are suggested as important contributors to clinical manifestation [12].

Extensive SGPV-associated erythrophagocytosis has only been described in cases of severe SGPVD with high virus loads [7]. Preliminary investigations give no proof of viremia during the course of the disease [11]. Erythrophagocytosis is reported in koi (*Cyprinus carpio*) with KSD [21] and in salmon infected with infectious salmon anaemia virus [22]. In the latter case, this is observed to a lesser extent than during SGPVD and with a distribution less restricted to haematopoietic organs. The pathogenesis leading to erythrophagocytosis in SGPVD is not well understood. In mammals, erythrophagocytosis caused by the large DNA virus, Epstein–Barr virus, was shown to be a result of viral interference with a number of immune response mechanisms [23]. Interestingly, dysregulation of the immune responses related to the Epstein–Barr virus reaches a tipping point on the proliferation of the virus to a remarkably high titre [24], and it is tempting to speculate whether a similar scenario contributes to severe outbreaks of SGPVD.

### 2.2. SGPV as a Contributor in Complex Gill Disease

The list of agents infecting salmon gills is long, and the pathogenicity of individual agents is poorly understood. A major obstacle in establishing controlled gill infection experiments is that most of the pathogens have not yet been cultured. ‘Complex gill disease’ or ‘complex gill disorder’ (CGD) are terms now used to describe gill disease manifestation in which the histopathological pattern is complex and several pathogens are suspected to be involved. The role of SGPV in CGD is not well understood, but the virus is sporadically detected in salmon with CGD, both in hatcheries [8] and following sea transfer [8,13,25]. In such cases, the SGPV signature in the form of apoptotic gill epithelial cells can easily be overlooked amongst the complexity of the histopathological changes seen in CGD. Therefore, a PCR approach targeting the most common and known agents is recommended when assessing gill diseases of salmon [13,25].

## 3. Diagnostics of SGPVD

The acute disease course and synchronicity with high mortality in some outbreaks is striking and can be mistaken as acute toxic events. Autopsy findings following both types of event are commonly non-specific. However, a histological assessment gives valuable clues in the case of SGPVD, as gill epithelial apoptosis is an almost pathognomonic finding [6,7,8,9,10,11]. Erythrophagocytosis, observed in some salmon suffering from SGPVD, is also a common finding in salmon with infectious salmon anaemia (ISA) [22] and other septicaemic diseases. Haemorrhagic smolt syndrome (HSS), a disease of uncertain aetiology affecting smolts, is associated with extensive bleeding and erythrophagocytosis in various organs [26]. In SGPVD, blood cell destruction appears to be localized to the spleen and kidney. In comparison, in ISA erythrophagocytosis may be seen in several organs, while in HSS, there is also widespread haemorrhage in the skeletal musculature, perivisceral fat, kidney, and heart [26].

For the diagnosis of infectious diseases, we strive to detect the causative agent using at least two biologically independent methods. This may, for example, involve the detection of the pathogen’s nucleic acids (e.g., with PCR or in situ hybridization) or proteins (e.g., with immunohistochemistry) in host tissues. In situ methods allow the assessment of the histological localisation of the agent, cell tropism, and characterisation of the associated lesions. A diagnosis of SGPVD is based mainly on clinical assessment and histopathology in combination with qPCR.

Although an immunohistochemistry method detecting the SGPV L1R protein (homologous to a surface protein in vaccinia) was successfully developed and used to confirm PCR detection and localize poxvirus within infected gill tissues [6,10], this method has not been validated for diagnostic use due to its low sensitivity. For the fowlpox virus (FPV), a distant relative of SGPV, a PCR amplification of a 578-bp fragment has been used as a diagnostic tool [27,28]. However, to arrive at an SGPVD diagnosis, clinical signs of the disease and characteristic pathological findings must be present in addition to a positive SGPV PCR. Since it is a DNA virus, a diagnostic PCR for SGPV can be based on the detection of either DNA or RNA. When directed at a single-copy DNA target, real-time PCR is indicative of the number of SGPV virions within the sample while RNA-based real-time PCR may be used to study viral gene expression. At the Norwegian Veterinary Institute, a DNA-based PCR approach is used to detect SGPV [6] and also for CEV, a DNA-based PCR is used [12,29]. These DNA-based PCR methods allow, to some extent, an estimation of the virus load, which is a strength in diagnostics, as it is so closely linked to disease severity and pathology. The in situ hybridisation of virus DNA or RNA has also succeeded in localising the poxvirus within salmon tissues (Figure 5) [11], and its validity for diagnostic purposes is currently being tested. Using these different methods in combination has confirmed a quantitative link between SPGV load, pathology, and disease manifestation [6,7,8,10,11].

## 4. SGPV Is an Epitheliotropic Virus

SGPV constitutes the deepest member of the Chordopoxvirinae family with a linear double-stranded DNA genome spanning ca. 242 kbp [6]. Although other salmonid species have only been investigated to a limited degree, the virus is so far exclusively associated with salmon, where it specifically infects epithelial cells of the gills and oral cavity [10]. Due to a lack of reproducible culture techniques, experimental SGPV infection trials currently rely on the use of SGPV-infected tissues as challenge material, which significantly hinders research on the pathogenesis of SGPVD. It is presumed that SGPV enters a new host by infecting the gill epithelium, where propagation of new virus particles takes place. Interestingly, despite the highly vascularized nature of the gills, they are the only organs showing high viral loads during acute infection and there is still no evidence of hematogenous spread of SGPV [11]. Generally, most viruses exploit numerous surface receptors and co-receptors of host cells to gain cellular access, but the aforementioned lack of culture techniques has again prevented studies on SGPV biology. Pentameric glycoprotein complex receptors that may facilitate viral infection of epithelial cells [30] should be explored for their possible contribution to SGPV attachment and entry into target host cells.

## 5. Epidemiology

SGPVD is not a notifiable disease in Norway or in the World Organization for Animal Health (WOAH) system, and there is no official control program for SGPVD in Norway, thus rendering the exact number of annual cases uncertain. However, extensive real-time PCR screening has shown SGPV to have a wide geographical distribution, with detections in Norway [6], the Faroe Islands [10], Scotland [10], Iceland, and Canada [31] collectively spanning all physiological life stages of farmed salmon [10,13,15]. SGPV is also regularly detected in wild salmon in Norway [32]. The investigation of diagnostic archival samples has furthermore demonstrated that SGPV has been present in salmon freshwater farms in Norway since at least 1995 [8].

### 5.1. A Discrete Trans-Atlantic Divide within the SGPV Population

A single SGPV whole genome sequence originating from the infected gills of Atlantic salmon in Norway in 2012 remains the only complete genome publicly available to date [6]. Partial genomes and/or genes from a few other SGPV strains have also been released [31,33]. The phylogenetic analysis of SGPV is based on the major capsid protein and DNA-directed RNA polymerase subunit beta genes. It involves strains collected on both sides of the northern Atlantic Ocean (i.e., including the single detection reported from Canada) [31] and indicated a discrete trans-Atlantic divide within the SGPV population (Figure 6) [33]. Notably, a recent meta-transcriptomic viral survey of freshwater fish in Australia identified a novel poxvirus in western carp-gudgeon (*Hypseleotris klunzingeri*), which together with SGPV (based on the DNA polymerase gene) forms a distinct fish-associated lineage within the Chordopoxvirinae, thus indicating virus-host co-divergence [34].

### 5.2. No Significant Genetic Distinction Was Found between SGPV Respectively Associated with Clinically Healthy Carriers and High-Mortality Outbreaks

The strains examined from the northeastern Atlantic region, from where the vast majority of SGPV reports derive, have thus far presented as phylogenetically conserved (Figure 6), but the introduction of high-resolution MLVA genotyping has shed some light on the epizootiology underlying SGPVD [33]. Importantly, however, using these assays no significant genetic distinction was discerned between SGPV samples collected respectively from clinically healthy carriers and from high-mortality outbreaks. This again points to the likely pivotal role of other factors relating to the host and/or environment in disease development.

### 5.3. Prevailing House Strains

The comparison of MLVA-profiles indicates the existence of SGPV ‘house strains’ prevailing and causing recurring outbreaks over several years within individual freshwater salmon hatcheries in Norway [33] (Figure 7). Eradicating SGPV from a facility may, therefore, be challenging and any small crevice left un-sanitized may represent a potential risk. This was illustrated in one farm where a vaccination robot was verified by MLVA as the source of SGPV reintroduction following an otherwise apparently successful sanitizing effort [35]. MLVA genotyping was further used to refute two cases of suspected vertical SGPV transmission. While ‘house strains’ may be detected in individual facilities, the MLVA profiles of SGPV samples from geographically disparate salmon hatcheries in Norway have indicated no overarching geographic patterns for the aquaculture industry as a whole.

### 5.4. Genetically Similar SGPV Strains Are Found in Geographically Linked Fjord Systems

Contrary to the situation in farmed salmon, SGPV-infected wild salmon returning to spawn in Norwegian rivers within connected fjord systems have shown a tendency towards carrying related SGPV strains [33]. The degree of strain conservation does not increase in individual river systems [33], however, which may indicate that transmission between wild salmon predominantly takes place in fjords and other coastal areas. Although the transmission of SGPV between farmed and wild salmon has not been reported to date, Norwegian law requires the disinfection of farm intake water from river systems housing populations of anadromous fish.

### 5.5. Surveillance of SGPV by Detection in Water

Currently, surveys of salmon for SGPV largely depend on the sampling of gills for histopathology, nucleic acid isolation, and amplification with qPCR. These methods have yielded valuable insights and revealed many infections, but they also have limitations. The difficulties involved in gill or swab sample acquisition from live fish tends to limit the feasibility and effectiveness of such surveys. Weli and co-authors, therefore, established an environmental RNA/DNA (eRNA/eDNA) detection-based methodology for surveillance of viruses in fish farms [36,37]. The method, involving the sampling of seawater from salmon hatcheries and farms, the concentration of the virus, and subsequent molecular detection (37, 38), has been adapted and used for the detection of SGPV in salmon in freshwater in hatcheries (unpublished). The assessment of SGPV in water samples in surveillance and monitoring programs may provide fish health inspectors with a continuous overview of viral loads in salmon hatcheries and farms, which could conceivably serve as an early warning system for the onset of SGPVD [9].

## 6. Host Immune Responses to SGPV Infection

Two studies have addressed the immunological responses to SGPV. One characterised gill transcriptomes during a field outbreak [7] while the other examined immunological responses in a controlled SGPV trial with cortisol injection [11]. In both studies, the histological examination of gills during the acute mortality phase of SGPVD revealed typical pathological destruction of the gill epithelium but no notable infiltration of immune cells into the gills.

### 6.1. Interferon-Regulated Genes and Genes Involved in Antigen Presentation Are Upregulated in SGPV-Infected Gills

In a transcriptome- and histology-based study involving a SGPVD outbreak in a hatchery, pathological changes and immune responses were explored. The samples were collected two weeks prior to, during, and two weeks after acute SGPVD mortality. The transcriptome analysis of gill samples showed that several interferon-stimulated genes (ISGs) were strongly upregulated in the acute phase, a response that lasted into the late phase [7]. The same response was also monitored with RT-qPCR in an in vitro controlled infection trial, in which the IFN-regulated antiviral genes Mx and ISG15 were strongly upregulated in gills throughout the infection course with a higher induction in the group suffering severe disease and mortality [38]. For other poxviruses, the blocking of IFN signaling is reported as one of the main counteracting mechanisms on the antiviral response [39], and SGPV infection appears to differ in this regard. Some of the upregulated ISGs encode proteins shown to directly affect the replication of other poxviruses in other host species, including ISG15 [40], Barrier to Autointegration Factor/BAF [41], sterile alpha motif domain-containing 9-like (SAMD9L) [42], and suppressor of cytokine signaling (SOCS)- 1 [43]. Although the transcription of antiviral genes is induced in salmon gills, the antiviral gene products may not be functional, a phenomenon also shown for other poxviruses [41,42,44]. The genes involved in antigen presentation were upregulated in the acute phase in gills [7], indicating that SGPV-infected cells may present viral proteins during infection. The downregulation of various genes associated with toxic defense mechanisms was also found in the gill transcriptome study during and after SGPVD, including genes linked to oxidative stress responses and xenobiotics mechanisms [7], which may render gills more sensitive to environmental impact.

### 6.2. Adaptive Immune Responses—A Role of Cytotoxic T-cells in SGPV Defence?

Both the transcriptome study from the field outbreak of SGPVD and the SGPV challenge experiment reported earlier [7,38] indicated a limited adaptive immune activation in the gills. In the transcriptome study [7], the gene encoding lymphocyte-associated Artemis like protein, known to be involved in V(D)J recombination for both B cell antibody- and T cell receptor- genes, was downregulated during SGPV infection. The same was true for the innate T-cell receptor (TCR) Fcγ gene and the associated cytokine IL-22, indicating a deficiency in mucosal T cell responses [7]. The chemokine CCL19 is involved in T-cell recruitment during viral infection and was shown to be strongly induced in gills infected with SGPV [7]. Further, in situ hybridisation investigations strongly suggest a role of CCL19 in the onset and maintenance of response to SGPV in the gill (Bøyum, S. Manuscript in preparation). Further investigations of cytotoxic T-cell responses in the gills and spleen of SGPV-infected salmon were performed at different time points during the experimental SGPV infection [38]. The results revealed that cytotoxic T-cell marker genes like CD8α (co-receptor for the T-cell receptor [TCR] on cytotoxic T cells), GzmA (one of the CD8+ cell killing mediators), and IFNγ (produced by activated cytotoxic T-cells) were all strongly induced in gills in fish with moderate SGPV levels and no disease. In contrast, fish injected with the stress hormone cortisol had low expression of these T-cell associated genes until directly prior to the acute SGPVD mortality phase. This suggests an apparent lack of cytotoxic T-cell recruitment to gills during the early phases of infection, perhaps related to the immunosuppressive effect of cortisol. Such a relation between stress and the recruitment of CD8+ cells to the infection site could partly explain the low numbers of inflammatory cells in gills infected with SGPV in the field study. In this case, prior stress events may very well have triggered the outbreak.

In humans, vaccination against smallpox using vaccinia virus generates long-lasting T cell responses to poxvirus antigens. Virus-specific CD4+ and CD8+ T cells can be detected several decades after a single smallpox vaccination. In that response, CD4 T cells were more stable and showed a smaller contraction between the peak effector and memory phase than that of the CD8 T cells [45]. In other poxviruses, including for example monkeypox (MPXV), cowpox (CPXV), and variola virus, B22 family proteins (encoded by *B22R* genes) were suggested to be responsible for virus virulence by suppressing the host T cell responses [46]. The deletion of genes encoding B22 proteins in MPXV has been shown to severely attenuate the virus and prevent lethal disease in rhesus monkeys [46]. In salmon, disease development and suppression of the mucosal T cell immune response after exposure to SGPV corresponded with an early expression of SGPV B22R genes in the gills already one day post-virus exposure. This suggests a similar immunosuppressive role for the SGPV B22 protein. While it was shown that the involvement of T cell subsets was required for protection against primary infection with ectromelia virus (the causative agent of mousepox) in mice, the elimination of the secondary infection was orchestrated by pox-neutralizing antibodies. However, expression analyses of immunoglobulin genes (*IgM* and *IgT*) in the gills and anterior kidney of salmon fish infected with SGPV showed no upregulation (unpublished). Thus, there is no primary evidence that salmon can develop a specific humoral immune response against SGPV. It is nevertheless worth mentioning that the samples analysed in our studies (from both natural outbreak and experimental infection) were collected no later than one month after the initial SGPV exposure. A later development of a specific antibody response cannot, therefore, be ruled out. Indeed, the limited adaptive immune response of salmon against SGPV reported so far in our studies could explain why previously SGPV positive fish groups may become reinfected following sea transfer. This is in agreement with the suggestion of Østevik et al. that previous SGPV infection does not protect against a reinfection [25].

## 7. Propagation of SGPV under In Vitro Conditions

Severe SGPVD with high associated mortalities and the lack of any specific anti-SGPV treatment, combine to represent one of the major health threats in Norwegian salmon farms and hatcheries. The development of antiviral treatments against SGPV will necessarily require a robust virus cultivation system. In a study performed by a Canadian group [31], the permissibility of Chinook salmon embryo cells (CHSE-214) to the North American (NA) variant of SGPV was proven. In that study, a tissue pool containing kidney, spleen, pyloric caeca–pancreas, and gills from fish infected with SPGV was inoculated in CHSE-214 cells. The cells displayed cytopathic effects (CPE) on day 21 post-inoculation. The lysate from these cells was found to be infective in non-exposed CHSE-214 cells. TEM and high throughput sequencing confirmed the filterable replicating agent in the lysate to be the NA SGPV. Our first attempt to propagate the Norwegian variant of SGPV in cell culture was performed in 2015 on the Atlantic salmon kidney (ASK) cell line (ATCC^®^ CRL-2747™) using salmon SGPV-positive gill tissue homogenates, as determined by qPCR and TEM. The infection attempt did not result in SGPV replication as assessed by qPCR, and no cytopathogenic effects (CPE) were observed. Another attempt was performed four years later using Atlantic salmon gill (ASG-10) cells [47] and CHSE-214, utilising SGPV-positive material concentrated from water samples obtained from an SGPV outbreak. The high viral load of the inoculant was confirmed using the virus concentration method developed by Weli et al., 2021 [37]. In this attempt, the effect of a different incubation temperature (4, 15, and 20 °C) and cortisol exposure on SGPV propagation was assessed. However, sequential qPCR analyses over four weeks following virus exposure showed no evidence of virus replication.

## 8. Conclusions

Infection with SGPV in salmon can manifest in different ways: from no disease at all, to high mortalities. It is also a contributor to complex gill disease. There is now a strong body of evidence indicating that SGPV load, clinical manifestation, and pathological changes in gills are closely linked. Stress as a predisposing factor for a severe outbreak of SGPVD has been described both from the field and in experimental challenge trials. The recognition and control of the factors leading to development of severe disease would be of great value. No genetic determinant has yet been identified that discriminates ‘virulent’ SGPV from ‘avirulent or less virulent’ SGPV. An in vitro cultivation of SGPV has not been successful, and this has hampered elementary studies of the virus and its role in the development of SGPVD. A deficiency in early T-cell recruitment to SGPV-infected gills has been associated with a more severe disease outcome. There are a number of predicted proteins of unknown function in the SGPV genome that remains to be characterised.

## 9. Burning Questions

How can a cultivation system be developed, and may gene editing solve this challenge?

Is the apparent lack of difference in virulence due to limited access to complete SGPV genomes, or does it merely point towards host or environmental factors as primary triggers for severe disease outbreaks?

Are T-cells responses in the early phase of infection the most important protective response?

Can identification of the function of SGPV-encoded proteins shed light on disease pathogenesis and the apparent immunosuppression following SGPVD?

Could the loss of muscular flexibility, skin redness and erythrophagocytosis seen in some salmon experiencing SGPVD indicate key pathophysiological processes of potential relevance for disease development?

## Figures and Tables

**Figure 1 viruses-14-02701-f001:**
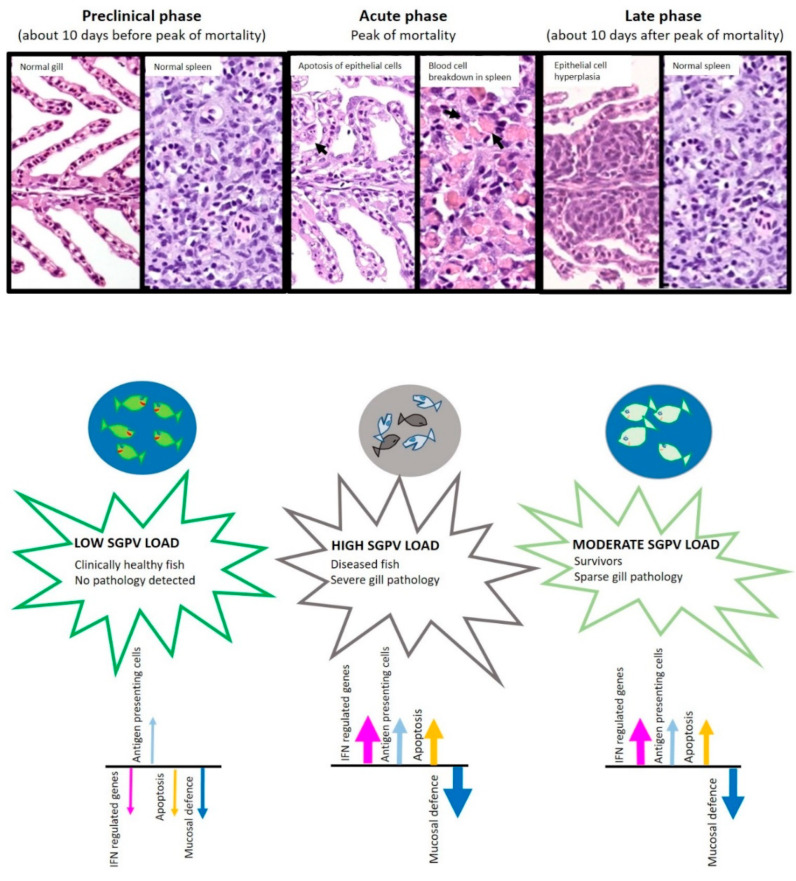
Salmon gill poxvirus disease course from the preclinical phase to the late phase. Histology images (top) show gills and spleens in the different phases of the disease. The lower panel indicates information gained from transcriptomic data from a field outbreak [7] with arrows illustrating gene regulation (up/down/arrow thickness indicates degree) associated with different biological processes in the gills at the different stages of disease. During the pre-clinical phase, the gills appear normal and with no histologically observable erythrophagocytosis in the spleen, the SGPV load is low, and gene regulation is not significantly affected. During the acute phase, extensive apoptosis of gill epithelial cells is seen and, in some cases, erythrophagocytosis in the spleen. High SGPV levels, upregulation of IFN-regulated genes, and downregulation of mucosal defence genes in the gills are also identified. In the later phase, sparse hyperplasia of gill epithelial cells can occasionally be seen, spleens appear normal, and SGPV levels are moderate. IFN regulated genes remain induced in the gills, and mucosal defence genes suppressed.

**Figure 2 viruses-14-02701-f002:**
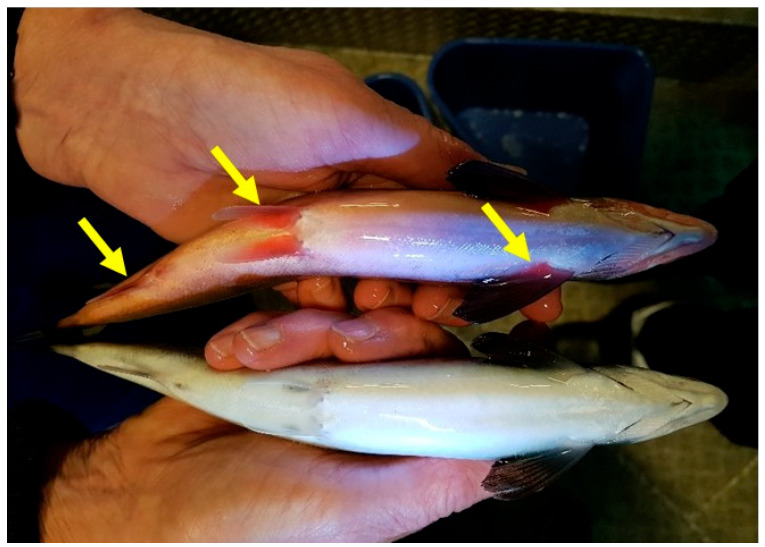
Salmon, both having suffered from acute salmon gill poxvirus disease (SGPVD). Note redness of the abdominal wall and fin of the upper fish (yellow arrows), a common finding in some individuals in SGPVD outbreaks.

**Figure 3 viruses-14-02701-f003:**
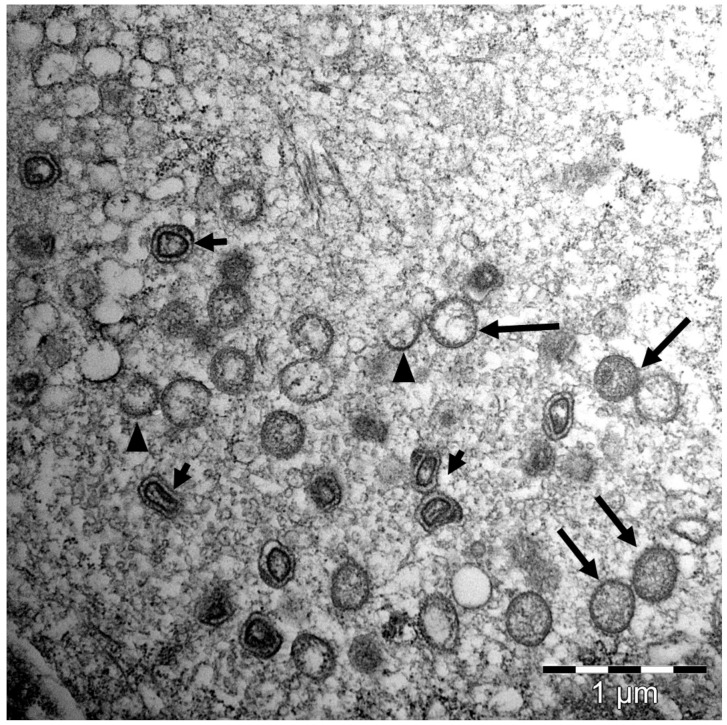
Transmission electron microscopy of the inside of a salmon gill epithelial cell infected with SGPV showing different stages of SGPV morphogenesis. Crescents (arrowheads), immature virions (large arrows), and mature virions (small arrows).

**Figure 4 viruses-14-02701-f004:**
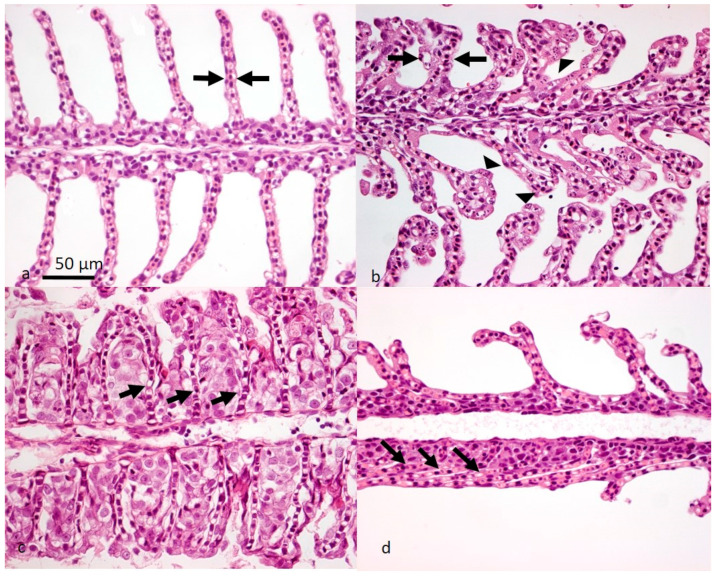
Three examples of SGPVD gill pathology leading to severely reduced functional respiratory surface area. (**a**) Normal gill histology shows thin lamellae (arrow) with normal epithelial cells narrowly separating blood and water. (**b**–**d**) SGPVD gill pathology i.e., (**b**) the extensive presence of apoptotic gill epithelial cells (arrowheads) containing SGPV, and lamella that have become thickened (arrows). (**c**) Foamy epithelial cells fill the normally water-filled inter-lamellar space (arrows indicate lamellar vessels), and (**d**) possibly detached apoptotic cells with associated loss of surface charge, leading to lamellar adhesion (arrows).

**Figure 5 viruses-14-02701-f005:**
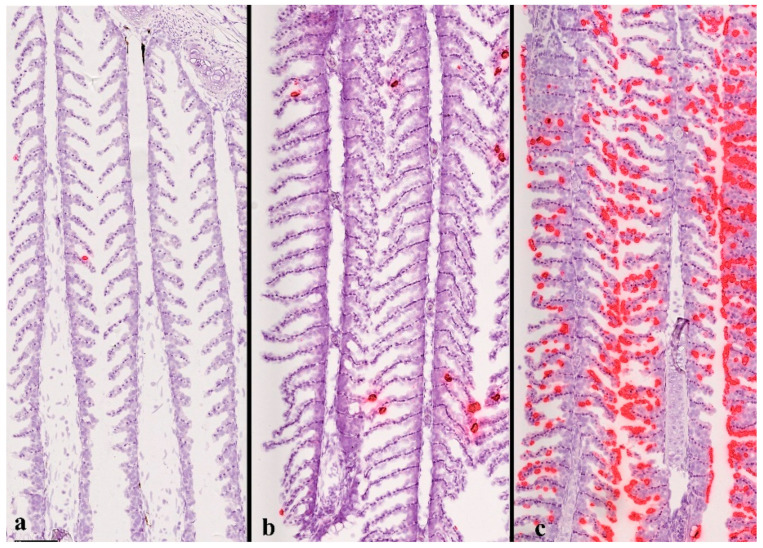
In situ hybridisation of SGPV D13L RNA in salmon gills infected with SGPV. Different stages of disease during an SGPV challenge experiment from (**a**) early, (**b**) mid infection course, and (**c**) at peak of fish mortality with a gradual increase in virus load (red staining). Scale bar 100 μm. Reproduced from Thoen et al. [11].

**Figure 6 viruses-14-02701-f006:**
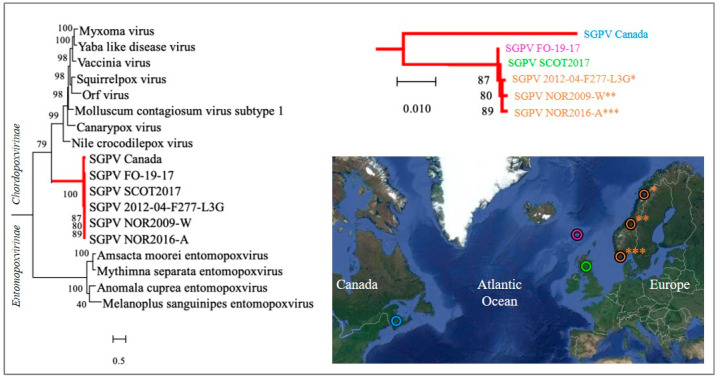
Maximum likelihood phylogeny of SGPV strains. SGPV phylogeny in relation to other poxviruses (left) and within SGPV (upper right). The map on the lower right shows the Atlantic origin of the included SGPV strains (see colours). Phylogenetic analysis was based on concatenated amino acid sequences from the major capsid protein and DNA-directed RNA polymerase subunit beta. Figure modified from Gulla et al. 2020 [33].

**Figure 7 viruses-14-02701-f007:**
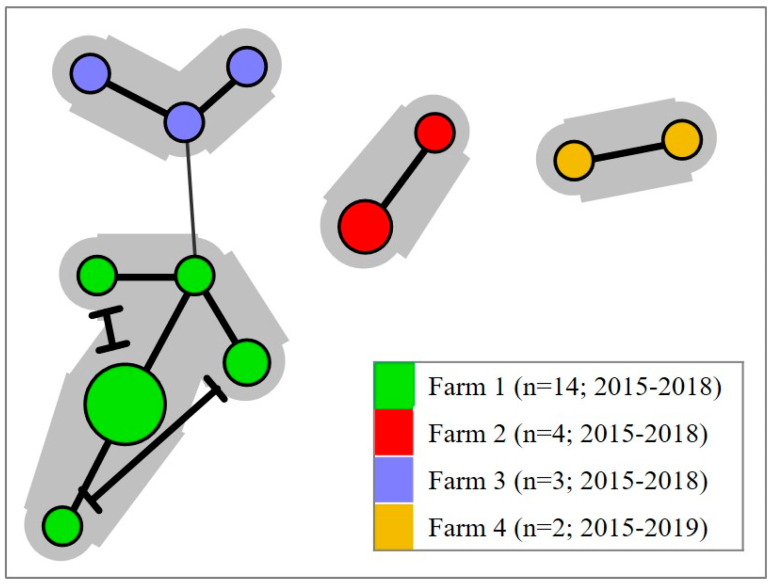
Farm–specific clustering of SGPV variants. Minimum spanning tree (created in BioNumerics v7.6.3) based on MLVA-profiles from 23 SGPV samples originating over several years (2015–2019) from four selected freshwater salmon hatcheries in Norway. Thick and thin node connections reflect, respectively, 7/8 and 6/8 shared VNTR loci, while connections of lower similarity are hidden. A clear tendency of farm-specific clustering can be observed over the sampling period(s).

## Data Availability

All data is referred to in the references.

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
