# Peer review of "Emergence of Salmon Gill Poxvirus"

_viruses, 2022, doi:10.3390/v14122701_

Round 1

Reviewer 1 Report

Dahle et al.

     This manuscript summarizes the current knowledge status, highlights factors related to disease manifestations of SGPV, explains how the host immune system can be affected, describes the dissemination of the virus, and shows how newly established molecular tools can help us towards a better understanding of SGPV and its pathogenesis. Overall, the research objective is clear and the approach is sound and methodical. Few minor comments/questions are as below to address before publication.

Q1- The language of the manuscript is perfect. However, my general impression is that several sentences are too long with so many conjunctions which can make reader confuse or misleading.

Q2- In 3. Diagnostics of SGPVD, you did not mention viral load as diagnostic, although figure 1 shows viral load severity according to disease progress?

L8-9- ‘….high mortality in Atlantic salmon presmolts..’. Is it not farmed Atlantic salmon that is mainly affected due to SGPV.

L22- 350 mil. Spell it for first time.

L24- ‘….and prior to sea transfer – in freshwater farms (1)’. As disease is more relevant to fresh water pre-smolt then it would be good to give mortality rate in freshwater as well.

L30- ‘Compromised gills will, therefore, affect the fish at many levels’. You mean levels as mentioned above?

L39-41- ‘and in some cases cause epithelial destruction and severe and acute respiratory disease, and in some fish, also signs of extensive blood cell breakdown’. Too many ands without commas.

Fig. 1- Too small histology images to note changes in gills. What do you mean by APC as shown in figure and even not mentioned in the figure legend?

L56-‘…. genes in gills.

L64-65- ‘….severe respiratory distress leading to acute mortalities with losses of up to 70% in the affected tank’. Is it 70% fish or biomass loss?

Figure 2- Shadow in the image on right side. Better to label the lesions with arrows.

Figure 3- Arrows on right side are in middle of two viral particles. Please adjust it.

Figure 4A was not mentioned in the text.

Figure 4D- arrows were too small and even not mentioned in the figure legend.

L128- KSD. Spell it for first time.

L161- ‘2.2. SGPV as a contributor in complex gill disease’. Use same size font.

L207- ‘These methods allow, to some extent..’. Which methods you mean?

Figure 5- Use arrows at least in a & b to point out red labelling.

Figure 6- Map image is too small to see SGPV strains in Norway. Use same size font for figure legend.

Figure 7- ‘(see legend)’. Which legend are you mentioning here?

L328-29- ‘The method which involved sampling of seawater from salmon hatcheries and farms,…’. How about method for freshwater which is more relevant for SGPVD?

L381- sonset. Is it correct?

L434- ‘salmon (apparently healthy) was inoculated in the CHSE-214 cells….’. Although you explained below that fish had SGPVD but apparently healthy will confuse reader that healthy fish carries the virus. Please re-write it.

L452- Conclusions and burning questions

Author Response

Rev 1

 This manuscript summarizes the current knowledge status, highlights factors related to disease manifestations of SGPV, explains how the host immune system can be affected, describes the dissemination of the virus, and shows how newly established molecular tools can help us towards a better understanding of SGPV and its pathogenesis. Overall, the research objective is clear and the approach is sound and methodical. Few minor comments/questions are as below to address before publication.

Q1- The language of the manuscript is perfect. However, my general impression is that several sentences are too long with so many conjunctions which can make reader confuse or misleading.

The language of the manuscript has been worked through by an English speaking colleague.   

Q2- In 3. Diagnostics of SGPVD, you did not mention viral load as diagnostic, although figure 1 shows viral load severity according to disease progress?

This is now clearly stated in the new version of the manuscript (lines 231 – 232).

L8-9- ‘….high mortality in Atlantic salmon presmolts..’. Is it not farmed Atlantic salmon that is mainly affected due to SGPV.

This is now specified in the text (line 9)

L22- 350 mil. Spell it for first time.

This is now corrected in the text (line 23)

L24- ‘….and prior to sea transfer – in freshwater farms (1)’. As disease is more relevant to fresh water pre-smolt then it would be good to give mortality rate in freshwater as well.

This is now stated in the text (line 26)

L30- ‘Compromised gills will, therefore, affect the fish at many levels’. You mean levels as mentioned above?

This is now re-written (lines 35 - 38).

L39-41- ‘and in some cases cause epithelial destruction and severe and acute respiratory disease, and in some fish, also signs of extensive blood cell breakdown’. Too many ands without commas.

This is now fixed in the text (line 50)

Fig. 1- Too small histology images to note changes in gills. What do you mean by APC as shown in figure and even not mentioned in the figure legend?

This is now stated clearly in the figure. The size of the figure was also increased.

L56-‘…. genes in gills.

This is now stated (line 69)

L64-65- ‘….severe respiratory distress leading to acute mortalities with losses of up to 70% in the affected tank’. Is it 70% fish or biomass loss?

This is now specified in the text (line 76)

Figure 2- Shadow in the image on right side. Better to label the lesions with arrows.

The lesion is now labelled with arrows.

Figure 3- Arrows on right side are in middle of two viral particles. Please adjust it.

This is now adjusted as recommended.

Figure 4A was not mentioned in the text.

The figure is now referred to in the text as ‘Figure 4’ Panels’ specifications were removed.

Figure 4D- arrows were too small and even not mentioned in the figure legend.

This is now fixed in both figure and caption.

L128- KSD. Spell it for first time.

This is now fixed in the text (line 94)

L161- ‘2.2. SGPV as a contributor in complex gill disease’. Use same size font.

This is now fixed (line 181)

L207- ‘These methods allow, to some extent..’. Which methods you mean?

This is now fixed in the text (line 231)

Figure 5- Use arrows at least in a & b to point out red labelling.

As the red staining intensity is distinguishable between the different panels, we prefer to leave the figure as it is right now with the virus load being referred to as (red staining).

Figure 6- Map image is too small to see SGPV strains in Norway. Use same size font for figure legend.

The image size and figure legend font are now djusted.

Figure 7- ‘(see legend)’. Which legend are you mentioning here?

This is now removed.

L328-29- ‘The method which involved sampling of seawater from salmon hatcheries and farms,…’. How about method for freshwater which is more relevant for SGPVD?

The method has been adapted for the SGPV detection in freshwater, and this is now stated in the text (lines 365 - 366)

L381- sonset. Is it correct?

The typo is now fixed in the text (line 420)

L434- ‘salmon (apparently healthy) was inoculated in the CHSE-214 cells….’. Although you explained below that fish had SGPVD but apparently healthy will confuse reader that healthy fish carries the virus. Please re-write it.

The sentence was re-written (line 477 - 478)

L452- Conclusions and burning questions

This section is now divided into two sections

Reviewer 2 Report

11.   “Farming of Atlantic salmon (hereafter salmon) in Norway is a huge industry, with 21 about 350 mill animals put to sea in 2021”- repharse the sentence

22. Autor can put the information of SGPV under the umbrella of level 1, Level 2 and level 3 diagnosis (OIE diagnosis).

33.   Line 175: What you mean by “some SGPVD” clarify

44.  Line 177: “histological assessments give valuable clues in the case of SGPVD as gill epithelial apoptosis is nearly pathognomonic” cite the reference

5.5.    Line 187: “In the diagnostics of infectious diseases” Change to ” In the diagnosis of infectious diseases”

66.  “SGPV isolation and detection” Change to propagation of  SGPV inder invitro condition”

77.    Section: SGPV isolation and detection: explained that, SGPV was not able to grow in invitro condition. Please mention the reason for the same.

88.    Review should have “field level diagnosis method” “non-invasive methods” “detection of virus in water”  the part

99.    The transcriptomic data can come under the section of NGS technique

110   As I mentioned in comment two- the information in the manuscript is good, but, it needs to be rewritten as per the OIE protocols. This will help readers to understand the concept well.

1 11. Conclusions and burning questions:: separate section for conclusion and burning questions

Author Response

Rev 2

 “Farming of Atlantic salmon (hereafter salmon) in Norway is a huge industry, with 21 about 350 mill animals put to sea in 2021”- repharse the sentence

This sentence is now re-written (lines 23 - 25)

Autor can put the information of SGPV under the umbrella of level 1, Level 2 and level 3 diagnosis (OIE diagnosis).

Thank you for the relevant suggestion to rewrite according to OIE protocol. However this manuscript in not describing specific protocol, rather an overview of research work on this new important fish pathogen. In the near future, when sampling regimes, disease scoring and other parameters, are optimized, we will be willing to write protocol according to OIE. We do think that, when writing OIE protocol, that it is important to have a complete picture of a topic before giving recommendation. We prefer to keep this in the present form to avoid repetitions and to restructure the manuscript

Line 175: What you mean by “some SGPVD” clarify

This is changed in the text (line 195)

 Line 177: “histological assessments give valuable clues in the case of SGPVD as gill epithelial apoptosis is nearly pathognomonic” cite the reference

The sentence is now properly cited (line 199)

Line 187: “In the diagnostics of infectious diseases” Change to ” In the diagnosis of infectious diseases”

This is now changed (line 208)

 “SGPV isolation and detection” Change to propagation of  SGPV inder invitro condition”

This is now changed as for the reviewer’s recommendation (line 467)

Section: SGPV isolation and detection: explained that, SGPV was not able to grow in invitro condition. Please mention the reason for the same.

Although we are doing our best right now to figure this out, no solid answer, justification or even a suggestion can be given. Anyway, this is also the case with different fish pathogens, so SGPV is not a special case here.

Review should have “field level diagnosis method” “non-invasive methods” “detection of virus in water”  the part

Of course, we would like to have a section on “field level diagnosis method” “non-invasive methods” “detection of virus in water”. However, data and publication available on non-invasive method for detection of pox in water is limited to date, and as such we will like to have more data from the field and hatcheries, before expanding the writing.

The transcriptomic data can come under the section of NGS technique

As some of the results obtained in the transcriptomic data is quite relevant to the different parts of the review, we believe that distributing them among the sections will help the different kinds of reader to get the benefit from reading the sections they are interested the most in. For example, people with immunology background might not be interested in reading the NGS part of the review.

 As I mentioned in comment two- the information in the manuscript is good, but, it needs to be rewritten as per the OIE protocols. This will help readers to understand the concept well.

Thank you for the relevant suggestion to rewrite according to OIE protocol. However this manuscript in not describing specific protocol, rather an overview of research work on this new important fish pathogen. In the near future, when sampling regimes, disease scoring and other parameters, are optimized, we will be willing to write protocol according to OIE. We do think that, when writing OIE protocol, that it is important to have a complete picture of a topic before giving recommendation. We prefer to keep this in the present form to avoid repetitions and to restructure the manuscript

 Conclusions and burning questions:: separate section for conclusion and burning questions

This is now written as recommended by the reviewer.
